# Factors contributing to differences in physical activity levels in (pre)frail older adults living in rural areas of China

Xin Zhang[1,2]*, Xiaoping Zheng[3], Hans Hobbelen[4,5], Barbara van Munster[6], Qian Tong[7], Tianzhuo Yu[2], Feng Li[2], Claudine J.C. Lamoth[1]

1 Department of Human Movement Sciences, University of Groningen, University Medical Center Groningen, Groningen, The Netherlands, 2 School of Nursing, Jilin University, Changchun, Jilin, China, 3 Department of Sports Science and Physical Education, The Chinese University of Hong Kong, Hong Kong, China, 4 Research Group Healthy Ageing, Allied Health Care and Nursing, Hanze University of Applied Sciences, Groningen, The Netherlands, 5 Department of General Practice and Elderly Care Medicine, University of Groningen, University Medical Center Groningen, Groningen, The Netherlands, 6 University Center for Geriatric Medicine, University Medical Center Groningen, University of Groningen, Groningen, The Netherlands, 7 Department of Cardiology, Bethune First Hospital of Jilin University, Changchun, Jilin, China

* x.zhang01@umcg.nl

**Peer Review History:** PLOS recognizes the benefiYts of transparency in the peer review process; therefore, we enable the publication of all of the content of peer review and author responses alongside final, published articles. The editorial history of this article is available here: https://doi.org/10.1371/journal.pone.0335607

## Abstract

### Introduction

Physical Activity (PA) is essential for enhancing the physical function of pre-frail and frail older adults. However, among this group, PA-levels vary significantly. Identifying the factors contributing to these differences could support tailored PA interventions. This study aims to examine factors associated with physical activity levels among pre-frail and frail older adults in rural China.

### Methods

This is a cross-sectional study. A total of 284 (pre)frail older adults (aged ≥60 years) were included from ten rural healthcare centers in Northeast China. Participants were categorized into low-moderate and high physical activity groups assessed using the Short Form International Physical Activity Questionnaire. Four-dimensional data were collected, including demographics, health behaviors, objective physical performance measures, and self-reported perceived health profiles. Extreme Gradient Boosting (XGBoost), a machine learning algorithm, was employed for binary classification (low-moderate vs. high physical activity). Model performance was assessed using the area under the receiver operating characteristic curve (AUC), accuracy, sensitivity, specificity, precision, and F1-score. To enhance interpretability, SHapley Additive exPlanations (SHAP) were utilized to identify key predictive variables.

**Data availability statement:** All relevant data are within the paper and its Supporting Information files.

**Funding:** This study was financially supported by Rijksuniversiteit Groningen (https://www.rug.nl) in the form of a grant (P312319) received by XZ. This study was also financially supported by the China Scholarship Council in the form of an award (202206170103) received by XZ.

**Competing interests:** The authors have declared that no competing interests exist.

## Results

Mean age of participants was 70 years (59% female, 86% farmers). The low-moderate group averaged 1,187 MET/week, while the high physical activity group reached 8,162 MET/week. Physical performance tests showed significantly better scores in the high PA group. The XGBoost model achieved 82.4% accuracy (AUC: 0.769, specificity: 90%, sensitivity: 63%). SHAP analysis revealed that self-reported social support, general health, ambulation, and physical performance measures were the most important factors.

## Conclusion

The high physical activity group demonstrated better physical function than the low-moderate physical activity group; though, both groups showed poorer physical function compared to the general older population. Self-reported health perceptions and social support significantly correlated with physical activity levels. Addressing these factors through targeted interventions—including community-based social support programs and structured mobility-enhancing exercises—may contribute to improved health outcomes and enhanced quality of life in this population.

## 1. Introduction

China's population is aging rapidly. It is predicted that by 2050, the number of older adults over 65 will rise to 26.1% of the total population in China [1]. As highlighted by a recent national survey in China, over 70% of older adults live in rural areas [2], with rural regions experiencing a disproportionately high burden of aging due to urban migration of younger populations. Compared to older individuals living in urban areas, older adults in rural China often face limited healthcare resources, lower socioeconomic status, and increased vulnerability to age-related health conditions [3,4].

Among older adults, one of the most common clinical syndromes is frailty, especially in northern China [4–6]. Frailty is a multicomponent condition, associated with increased dependency and a heightened risk of mortality [7]. Fried Phenotype is one of the most widely used frailty frameworks to assess physical frailty and operationalizes frailty based on five criteria: unintentional weight loss, muscle weakness (low grip strength), exhaustion, slow walking speed, and low physical activity [8].

Research indicates significant disparities in physical activity levels between rural and urban environments, with older individuals in rural areas exhibiting higher levels of physical activity. The international collaborative study of cardiovascular disease in Asia found that among individuals aged 35–75, 78.1% of rural residents engaged in at least 30 minutes of moderate-to-vigorous physical activity daily, compared to just 21.8% of urban residents [9]. Similar trends have been observed in other studies. A cross-sectional study in the Tarragona province of Spain reported that older adults in rural areas engaged in higher-intensity physical activities (76%) compared to those in urban areas (65%) [10]. Likewise, research in Portugal found that rural older

residents scored higher in health-related quality of life and functional fitness than their urban counterparts [11]. These disparities can be attributed to various factors, including geographic characteristics, urbanization, mechanization of daily tasks, limited free time, and increased reliance on vehicles in urban settings.

Engaging in moderate to vigorous physical activity has shown benefits for (pre)frail older adults, improving body functions and potentially influencing other components of frailty [2,12–14]. Physical decline is just one component of frailty. Frailty is a multidimensional condition influenced by multiple interrelated domains, including nutrition, cognition, psychological well-being, self-perceived health and social support [15,16]. Physical activity can serve as a crucial mediator in preventing (pre)frailty by positively influencing the above interrelated domains. As indicated by the Biopsychosocial Model of Health, both objective health measures and self-perceived health significantly influence physical activity [17]. While objective factors such as grip strength and walking speed establish physical capacity, self-perceived health including self-rated health, self-efficacy, and social support, play a critical role in motivating and sustaining activity [18,19]. A growing body of evidence supports the integration of these dimensions to better understand physical activity behavior within a holistic biopsychosocial framework, especially in understudied populations [20,21]. Self-reported factors, reflecting individuals' perceptions and beliefs about their health and well-being, have shown strong associations with physical activity levels. Higher self-perceived quality of life and greater social support have been linked to increased levels of physical activity [12,13,16,22,23]. While studies have addressed relationships between various individual health factors and physical activity among older adults using regression approaches, the integration of biological, psychological, and social domains and assessment of interactions among these factors will provide a more comprehensive picture of the relation between physical activity and different health domains.

This study aims to get insight into the interaction of different health domains on physical activity levels in older adults living in rural areas of China who have been identified as physically (pre)frail. Based on the biopsychosocial model of health, we hypothesize that self-perceived health measures will demonstrate associations with physical activity levels among pre-frail and frail older adults, and that the interaction between psychological, biological, and social factors will explain a significantly greater proportion of variance in physical activity than any single domain.

## 2. Methods

### 2.1 Study design

The study has been conducted in accordance with the principles expressed in the Declaration of Helsinki and approved by the Human Research Ethics Committee of the School of Nursing, Jilin University (Project-ID 2022082905).

In this observational study, potential participants were recruited from the rural Primary Healthcare Centre (PHC) by 10 local general practitioners from October 1 to November 15 of 2022. The general practitioner introduced the project to the potential participants in advance, then, scheduled an appointment with those who agreed to participate and signed the informed consent form. Considering the seasonal farming and inconvenience of transportation in the rural area, measurements of participants in the same village were scheduled for the same day and conducted at the local PHC. The research group has a long-term collaboration with these rural PHC, and a consensus on recruitment criteria was reached before the start of the project. Considering the low education level of most participants, all questionnaires and self-reported measurements were conducted in a one-on-one question-and-answer format and recorded by the investigators. Investigators were licensed health practitioners and trained to interview participants before the measurements.

### 2.2 Population

A total of 367 participants older than 60 have been recruited from 10 villages in rural areas of northeast China. Inclusion criteria for participation were, frailty score >0 according to the Fried Phenotype [8]; be able to walk independently without or with a walking stick for at least 3 meters; be able to understand Mandarin; capacity of consent. Exclusion criteria were, pain and severe mobility disability; unstable or persistent serious cardiovascular or respiratory system diseases (e.g.,

stage IV heart failure and COPD); Inability to communicate (e.g., severe hearing, vision, cognitive and language judged by primary health care practitioner). All participants provided a signed written informed consent before inclusion.

## 2.3 Assessment instruments

*Physical activity (PA) level*, was examined by the Short Form International Physical Activity Questionnaire (IPAQ-SF, Chinese version), which collects information on the frequency and duration (i.e., number of days and average time per day) spent in vigorous physical activity, moderate physical activity, walking activity, and sedentary behavior in the last seven consecutive days [24].Participants were labelled as having low to moderate and high physical activity intensities based on the MET (Metabolic equivalent – multiples of the resting metabolic rate) calculated according to the guidelines for interpreting the IPAQ-SF. High physical activity intensity is defined as at least 3 days achieving a minimum of 1,500 MET-minutes/week OR 7 or more days of any combination of walking, moderate-intensity or vigorous-intensity activities achieving a minimum of 3,000 MET-minutes/week. MET value is calculated as $8 \times$ high physical activity (min/week) + $4 \times$ moderate physical activity (min./week) + $3.3 \times$ walking (min./week) [25].

Four groups of variables were collected, demographic characteristics, health behavior, objective health performance, and perceived health profile (Fig 1a), and for details of the assessment instruments refer to the Supporting Information (Supplementary Material-Appendixes).

*Demographic and personal characteristics* were obtained from the general practitioners directly, including age, Waist Hip Rate (WHR), Body Mass Index (BMI), gender, occupation, education, income, marital status, living status, and cognitive impairment.

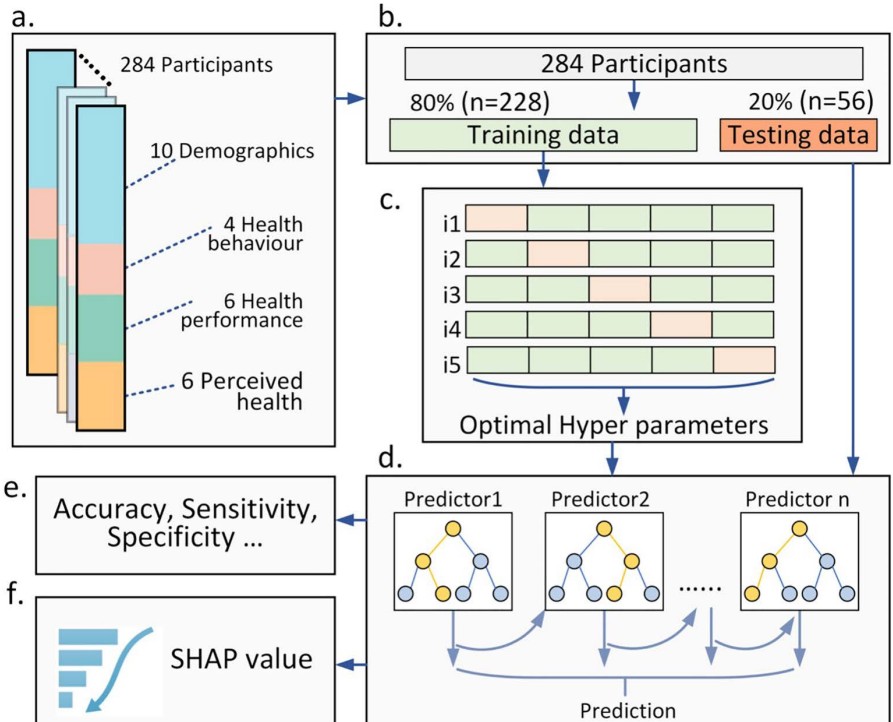

**Fig 1. The data processing and analysis pipeline.** (a) Data formulation; (b) Data splitting; (c) Determining optimal hyperparameters based on 5-fold cross validation; (d) XGBoost model training; (e) Model performance evaluation; f. Feature importance analysis.

*Health behavior variables* were collected by simplified "yes" or "no" questions, including smoking, drinking, daily intake of fibre, and daily intake of protein.

*Objective health performance variables* were calculated based on related physical tests or recorded from medical health records, including grip strength (GP), walking speed (WS) (3-meter walking test), time of the Time Up & Go test (TUG), total score of Short Physical Performance Battery (SPPB) [26], Mini-Mental State Examination (MMSE) [27], and number of comorbidities as recorded in the health records.

Regarding *perceived health profile variables*, the score of the General Anxiety Disorder-7 (GAD-7) [28], Patient Health Questionnaire-9 (PHQ-9) [29], and Social Support Rating Scale (SSRS) were obtained. The SSRS evaluates three key areas of social support: objective support, the actual support received (e.g., financial or material help), subjective support – how much emotional and psychological support a person feels they receive from others. Finally, it examines support utilization, which reflects how well and how frequently an individual makes use of their available support networks [30]. Subjective health and perceived functional strength and ambulation were assessed by three questions: "In general, how would you rate your health?", "Do you have any difficulty climbing a one floor stair or a 10-meter slope without resting and without assistance?", "Do you have any difficulty walking 500 meters alone without assistance?".

## 2.4 Data analysis

Statistical analysis of differences of individual variables between two groups was performed with Chi-Square tests for categorical variables and Mann-Whitney U tests for physical assessments, as the data were not normally distributed, $p < 0.05$ was considered statistically significant (Fig 2).

XGBoost (eXtreme Gradient Boosting) was used for this study due to its ability to handle imbalanced data, capture complex non-linear relationships, and provide feature importance insights, making it well-suited for binary classification tasks [31–33]. XGBoost can handle continuous, categorical, and binary variables without normalization because it uses decision trees, which split data based on relative ordering rather than absolute magnitude. All analysis were performed in R-studio (R 4.1.0 `xgboost` package version 1.4.1.1). The dependent variable (label) physical activity intensity was binarized to class 0 for low to moderate physical intensity and class 1 for high physical activity intensity level. Independent variables (input features were the 26 variables from the assessments of which 15 were binary scored variables (yes/no)).

**2.4.1 Architecture and pipeline of XGBoost.** The dataset was split into training (80%) and testing sets using stratified sampling to maintain balanced representation of the binary physical activity intensity classes. The data of the remaining 20% of participants were used as the testing dataset (Fig 1b). To address class imbalance (training data set N = 68 class 0 vs N = 160 class 1), class weights were calculated based on the ratio of negative to positive cases in the training dataset. This weight was used to balance the loss function during model training [34]. In this study, the high physical activity intensity group (class 1) was considered as the positive case and the low-moderate physical activity class the negative case.

A hyperparameter tuning process was conducted to optimize XGBoost using the training dataset. A grid search approach was employed to explore potential hyperparameter combinations, including learning rate (eta), maximum tree depth, number of boosting rounds, gamma, column sample by tree, minimum child weight, and subsampling ratio. Details has been provided in the supporting information (Supplementary Material-Appendixes).

To prevent overfitting and enhance model generalizability, a 5-fold cross-validation was applied during the validation phase. The training dataset was split into 5 folds, where each fold was used as validation while the remaining four folds were used to train the model with the selected hyperparameter combinations (Fig 1c). Mean squared error (MSE) was used to assess model performance during validation, and the average MSE across the 5 iterations represented the overall performance of selected hyperparameter combination. After identifying the best hyperparameters, the final model was retrained using the entire training dataset. Subsequently, the testing dataset was used to evaluate the generalizability of the model (Fig 1d).

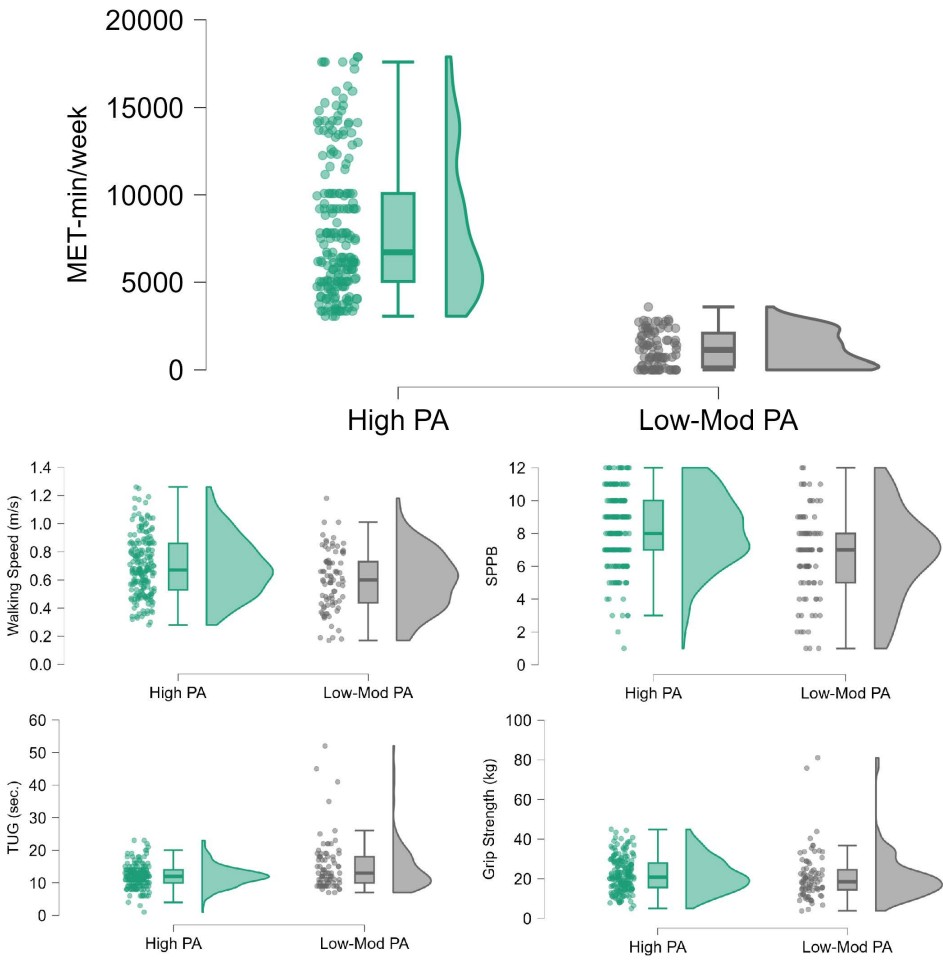

**Fig 2. Difference of overall MET values and variables derived from physical performance tests between low-moderate and high PA levels group.** Dots represent individual subjects, the boxplots depict the median (central horizontal line) and interquartile range (box), while the mean is indicated by a diamond symbol. The right portion of each boxplot displays the full data distribution for the respective group. PA = Physical Activity; MET = Metabolic Equivalent of Task; SPPB = Short Physical Performance Batter; TUG = Timed Up & Go test.

**2.4.2 Model performance assessment.** The overall performance of the XGBoost model was evaluated by the area under the receiver operating characteristic (ROC) curve (AUC). In addition, accuracy, sensitivity, specificity, precision, F1-score, were calculated (Fig 1e). Correct predictions for class 1 (high physical activity intensity group) are true positives (TP) and for class 0 (low to moderate physical activity intensity group) are true negatives (TN), while incorrect predictions are false positives (FP) and false negatives (FN). Accuracy is the proportion of all correct predictions: (TP + TN)/ (TP + FP + TN + FN). Sensitivity represents the proportion of positive cases correctly identified: TP/ (TP + FN). Specificity refers to the proportion of negative cases correctly identified: TN/ (TN + FP). Precision is the ratio of correctly predicted positive cases to all predicted positive cases: TP/ (TP + FP). The F1-score is the harmonic mean of precision and sensitivity: (2 × precision × sensitivity)/ (precision + sensitivity).

**2.4.3 Feature importance.** To get insights into the contribution of individual features (variables) to the classification model, SHAP (SHapley Additive exPlanations) values were computed (using the R-packages: SHAPforxgboost and shapviz) (Fig 1f). SHAP derived from cooperative game theory, allocate importance to features based on their contributions to predictions [35]. SHAP values were calculated for the test dataset. SHAP calculates the importance

of a feature by comparing the model's predictions with and without that feature, considering all possible orders and combination of feature inclusion to ensure a fair comparison. Higher absolute SHAP values indicate a high contribution to the classification process. Large positive values indicate that a feature drives the prediction toward the positive class, while negative values indicate that the feature pushes toward the negative class. A SHAP Summary Plot is presented displaying feature importance and effect direction.

## 3. Results

### 3.1 General characteristics

Of the 367 recruited participants, 284 were included in the analysis based on inclusion and exclusion criteria. 69 participants were not eligible: 47 participants had a Fried score of zero (indicating no physical frailty), 9 had unstable diseases, and 13 participants were missing the TUG test. After inclusion, an additional 14 participants were excluded during data inspection because their reported values fell outside the possible range for the questionnaire or assessment, likely due to data entry errors.

The mean age of participants was $70 \pm 5.5$ years (range: 60–87 years), with 59% being female. The majority of participants were farmers with low income and educational levels. The low-moderate physical activity group had a mean MET per week of 1,186.78 (SD = 1,012.52) and the high physical activity group of 8,162.47 (SD = 4,050.45) (Fig 2).

### 3.2 Difference of observed variables between low-moderate and high PA group

Analysis revealed significant differences between low-moderate and high physical activity groups regarding occupation, with a higher proportion of farmers in the high physical activity group. Income levels also differed significantly between groups, with the high physical activity group reporting lower average monthly income. Lower smoking rates were observed in the high physical activity group (Table 1).

As illustrated in Fig 2 and Table 1, significant between-group differences were observed in measures derived from physical tests. The high physical activity group, demonstrated better performance in walking speed, grip strength, TUG and SPPB sores compared to the low-moderate physical activity group. The distribution of the data showed wider distribution of the high physical activity group at the higher scores in particular for SPPB, walking speed and grip strength.

Self-perceived health and social functioning scores were significantly different between groups with the exception of the GAD-7 score (Fig 3, Table 1), which was highly variable as indicated by the multimodal distribution of data in both groups.

### 3.3 Performance of XGBoost model

The XGBoost model demonstrated good classification performance with an accuracy of 82.4% (95% CI: 70–91%). The model achieved an AUC of 0.769, sensitivity of 63%, specificity of 90%, precision of 71%, and F1-score of 67%.

Analysis of feature importance using SHAP values (Fig 4) revealed that the SSRS, general health questionnaire, TUG, ambulation capacity, and SPPB were the most influential predictors. Poor self-reported general health was associated with classification into the low-moderate physical activity group, while good health corresponded with the high physical activity group. Similarly, the ability to walk continuously for 500 meters was predictive of classification into the high physical activity group. While the SPPB and SSRS scores showed significant contributions to the model's predictions, their distributions were more heterogeneous within groups (Fig 1 and Fig 2).

## 4. Discussion

This study aimed to examine different factors correlated with physical activity levels in (pre)frail older adults living in rural China. The results highlight significant differences between individuals with low-moderate and high physical activity levels,

**Table 1. Demographics and outcome variables of the two groups.**

| Variables (unit) | Low-Moderate PA | High PA | Statistic | |
|---|---|---|---|---|
| N | 84 | 200 | – | – |
| Age (y) | 70.5 (5.6) | 69.2 (5.4) | U = 1.07 | p = .380 |
| Gender Female vs Male | 58% | 60% | Fisher | p = .960 |
| Occupation Farmer vs. other | 74% | 91% | X² = 10.65 | p = .001 |
| Education < 6 year | 79% | 79% | Fisher | p = 1.000 |
| Income < 3000 (¥/m) | 90% | 96% | X² = 4.29 | p = .038 |
| Married vs other | 76% | 81% | Fisher | p = .449 |
| Living with others Yes | 85% | 90% | Fisher | p = .328 |
| Smoking No | 45% | 33% | X² = 4.15 | p = .042 |
| Drinking No | 77% | 79% | Fisher | p = .959 |
| Daily Fiber Yes | 82% | 87% | Fisher | p = .447 |
| Daily protein Yes | 43% | 52% | Fisher | p = .230 |
| Comorbidities <=5 | 82% | 92% | X² = 5.91 | p = .015 |
| Walk >500 m | 58% | 85% | X² = 22.70 | p < .001 |
| Can walk stairs (functional strength) | 57% | 79% | X² = 14.19 | p < .001 |
| Good general health | 35% | 65% | X² = 21.54 | p < .001 |
| No cognitive impairment | 54% | 69% | X² = 5.73 | p = .017 |
| WHR | 0.92 (0.06) | 0.94 (0.85) | U = 1.01 | p = .314 |
| BMI | 23.40 (4.00) | 24.65 (3.50) | U = 2.19 | p = .028 |
| GP (kg) | 19.64 (8.48) | 21.99 (8.51) | U = 1.87 | p = .030 |
| WS (m/s) | 0.60 (0.21) | 0.67 (0.21) | U = 3.28 | p = .001 |
| TUG (seconds) | 14.98 (7.91) | 12.06 (3.25) | U = 2.43 | p = .020 |
| SPPB - | 6.67 (2.66) | 8.20 (2.34) | U = 4.40 | p < .001 |
| MMSE | 19.67 (4.96) | 21.81 (4.90) | U = 3.16 | p = .002 |
| PHQ9 | 8.31 (5.43) | 6.67 (4.57) | U = 2.31 | p = .021 |
| GAD7 | 7.31 (5.70) | 6.88 (5.35) | U = 0.61 | p = .541 |
| SSRS | 32.18 (7.71) | 36.35 (7.39) | U = 4.12 | p < .001 |

PA = Physical Activity; WHR = Waist-Hip Ratio; BMI = Body Mass Index; GP = Grip strength; WS = Walking Speed; TUG = Timed-up-and-Go; SPPB = Short Physical Performance Battery; MMSE = Minimal Mental State Examination; PHQ9 = Patient Health Questionnaire; GAD7 = General Anxiety Disorder Questionnaire Score; SSRS = Social Support Rate Scale; X² = Chi-square test; U = Mann Withney U test.

with key determinants including social support, self-reported general health status, and functional mobility. These findings contribute to the growing body of literature on aging, frailty, and physical activity, particularly in rural populations [2,4].

One of the most prominent findings of this study was the strong association between self-reported health, social support, and physical activity engagement. Consistent with previous research individuals with better self-rated health were more likely to engage in high-intensity physical activity [13]. Self-reported health in this study was assessed using a single-item measure: "In general, how would you rate your health?" with response options of "good" or "bad". Functional health was evaluated based on the ability to walk 500 meters and ascend a 10-meter slope or a one-floor staircase. Poor functional health, has previously been linked to lower positive affect (enthusiasm, pride, interest) but does not necessarily correlate with negative affect (anxiety, distress, anger) [36]. In line with these findings, participants' GAD-7 scores in this study, which reflect negative affect, were not elevated. However, this represents an association rather than causation. These results suggest that interventions targeting frailty should focus not only on improving physical function, but also incorporate strategies to enhance psychological well-being and promote positive affect.

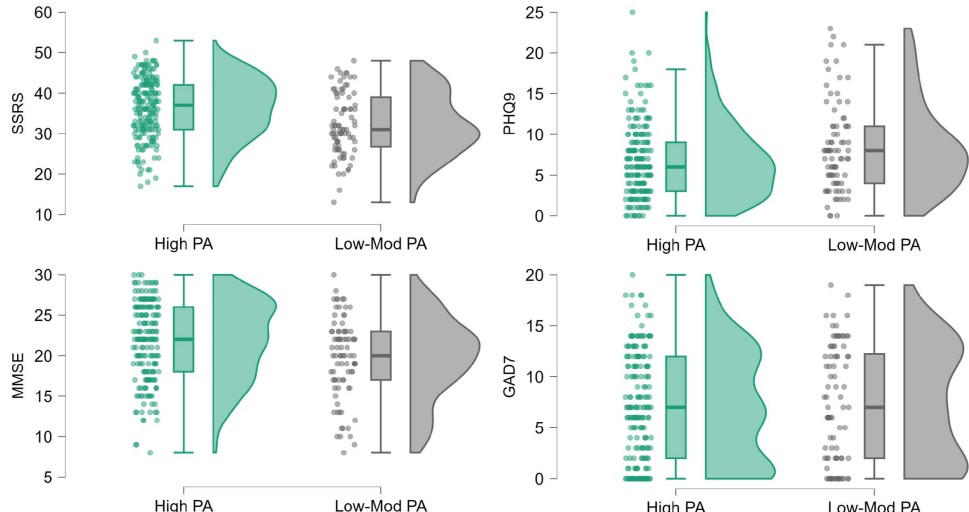

**Fig 3. Cognitive test and self-perceived health and social functioning presented for the low-moderate and high PA levels group.** Dots represent individual subjects, the boxplots depict the median (central horizontal line) and interquartile range (box), while the mean is indicated by a diamond symbol. The right portion of each boxplot displays the full data distribution for the respective group. PA = Physical Activity; SSRS = Social Support Rating Scale; PHQ9 = Physical Health Questionnaire-9; MMSE = Mini Mental State Examination; GAD7 = General Anxiety Disorder-7.

The role of social support, as measured by the SSRS, further underscores the psychosocial dimensions of frailty. Social networks and perceived emotional support provide motivation, assistance, and opportunities for engagement in physical activity [14], reinforcing the necessity of integrating social interventions into physical activity programs for (pre)frail older adults. The overlap between groups further indicates that social support alone is rarely sufficient to ensure high levels of physical activity. Success of interventions targeting social support depends on simultaneously addressing other individual specific barriers, such as functional limitations or perceived health status.

All participants in this study were classified as physically pre-frail and frail according to the Fried frailty criteria, with the majority exhibiting slow gait speed (99.3%) and low grip strength (59.2%). Mobility-related physical performance measures were significantly lower in the low-to-moderate physical activity group compared to the high physical activity group. However, normative data provided by Mayhew et al. indicates that even participants in the high physical activity group fell within the lowest 5th percentile for gait speed, grip strength, and TUG performance relative to age-matched counterparts [37]. The observed average gait speed of 0.6 m/sec is below the commonly reported cut-off of <0.8 m/sec for identifying clinically relevant slowness associated with frailty and sarcopenia [38]. Furthermore, the averaged SPPB scores ≤9, confirmed significant decline in walking ability, balance and muscle strength [39]. The IPAQ-SF assesses total physical activity without distinguishing between leisure-time physical activity (LTPA) and occupational physical activity (OPA), which may partly explain the observed paradox of high self-reported PA alongside poor physical performance. These findings addressed the necessity of considering the distinct roles of LTPA and OPA separately in future research, particularly in agricultural populations where occupational demands dominate daily activity patterns.

Despite these physical limitations, participants in the high physical activity group engaged in vigorous-intensity activity on at least three days per week, accumulating a minimum of 1,500 MET-minutes/week, or participated in a combination of walking, moderate-intensity, and vigorous-intensity activities for at least seven days per week, reaching a minimum of 3,000 MET-minutes/week. This may be attributed to the fact that the majority of participants were farmers, indicating physical activity as an integral part of their daily routines.

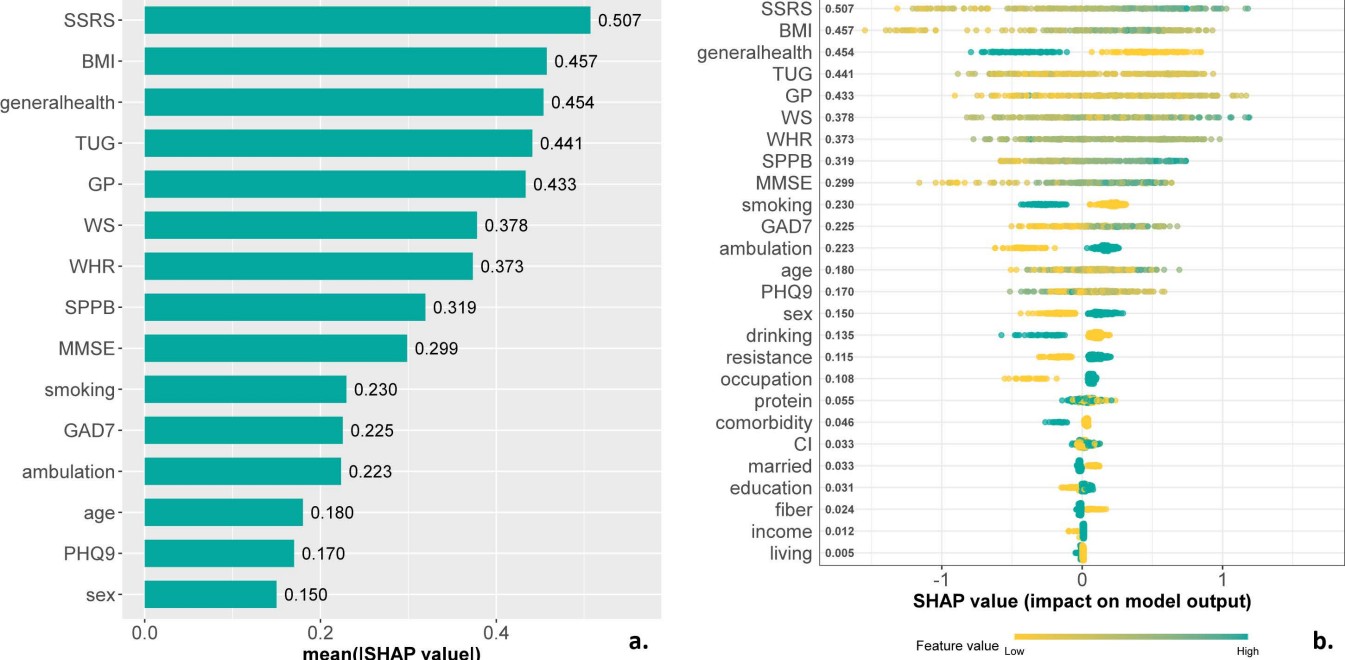

**Fig 4. The SHAP (Shapley Additive exPlanations) values of observed variables.** *Panel (a)* displays the mean absolute SHAP values, ranked by their relative contribution to the model; *Panel (b)* illustrates the direction and magnitude of influence for each input feature, with individual SHAP values represented as data points. Green circles denote higher feature values associated with increased likelihood of classification into the high physical activity group, while yellow dots indicate stronger predictive influence toward the low-moderate physical activity group. SPPB = Short Physical Performance Battery; TUG = Timed Up & Go test; SSRS = Social Support Rating Scale; PHQ9 = Physical Health Questionnaire-9; MMSE = Mini Mental State Examination; GAD7 = General Anxiety Disorder-7; WS = Walking Speed; BMI = Body Mass Index; WHR = Waist Hip Rate; GP = Grip Strength; CI = Cognitive Impairment.

The XGBoost model achieved a satisfactory classification accuracy of 82.4%. However, the model's accuracy suggests that unmeasured factors may also contribute to physical activity behaviors in this population. While the model exhibited high specificity (90%), moderate sensitivity (63%) indicates potential misclassification, particularly among individuals near the threshold between activity levels. Therefore, this model is positioned as a screening tool, rather than diagnostic, enabling healthcare providers to efficiently identify a high-risk subgroup for physical inactivity and prioritize limited resources toward those most in need. Future research should prioritize improving sensitivity, through methodological refinements, incorporating additional behavioral and environmental variables, to maintain acceptable specificity.

SHAP values revealed not only the importance of physical function (SPPB, TUG) and the ability to walk 500 meters unassisted, in facilitating greater participation in physical activity, but also highlighted the importance of the social support. A broader perspective on mobility and frailty—beyond the focus on physical frailty and physical performance—could offer deeper insights into the determinants of physical activity.

This perspective could incorporate the role of geographical life-space in shaping mobility behaviors. Life-space refers to the spatial extent within which individuals carry out their daily activities, encompassing the home environment, neighbourhood, and broader community interactions [40–42]. Mobility limitations, influenced by factors such as health status, demographic characteristics, and the built environment, significantly affect older adults' capacity for physical activity. Behaviors, such as pausing for rest during travel or the balance of time spent indoors versus outdoors, serve as indicators of the broader societal and geographical influences on physical activity [42]. In rural areas, where transportation infrastructure is often limited and communities are spatially dispersed, mobility constraints may restrict physical activity opportunities. However, the presence of strong social networks and expansive open landscapes may mitigate these limitations

by fostering active lifestyles [41,43]. Examining these behaviors can elucidate personal, social, and environmental determinants of physical activity among frail older adults. Understanding these contextual factors is crucial for developing targeted interventions that promote physical activity and mobility among older adults in rural areas.

It is worth noting that rural environments vary considerably across regions. For instance, Northeast China's rural areas characterized by severe continental monsoon climates with extended sub-zero winters, creating distinct living conditions compared to southern regions' milder year-round climate. These geographical variations, along with distinct cultural practices and social structures across different rural communities, may influence the generalizability of our findings. Nevertheless, our findings provide a valuable methodological framework that can inform future comparative studies across diverse rural settings.

The implications of these findings are substantial for interventions targeting frailty and physical inactivity in rural older adults. Given the significant role of social support, community-based programs emphasizing group activities, peer support, and family involvement may enhance participation rates. Considering that most of rural (pre)frail older adults already engaged in sufficient physical activity level, interventions for this population may need to focus on additional aspects, such as nutrition, which could provide greater health benefits [16,44]. For the PA intervention in this population, the primary objective is not to increase total PA volume, but to mitigate the detrimental effects of work-related movement and promote health-enhancing behaviors. Proposed strategies include health education to improve biomechanics and reduce injury risk, enhancing social participation during repetitive labor, and tailored community physical activity program during off-peak seasons designed to counteract physical strain and foster social engagement.

A limitation of this study pertains to the reliance on self-reported questionnaires for assessing physical activity, particularly the IPAQ-SF. This instrument was chosen because of several methodological and practical considerations specific to our study context in rural Northeast China. Although the IPAQ-SF is widely used to assess physical activity in the aging population, a recent systematic review reported that compared to objective physical activity measures with wearable sensors, the IPAQ-SF tends to overestimate activity levels [25]. The results reflect self-reported rather than objectively measured physical activity. While the associations identified between variables remain meaningful within the framework of this study, we recommend that future research incorporate device-based measures to further validate and extend these findings [45].

## 5. Conclusions

This study provides insights into the factors correlated with physical activity levels in (pre)frail older adults in rural China. The findings highlight the critical role of self-reported general health, social support, and functional mobility in determining physical activity levels. Addressing these factors through targeted interventions, including community-based social support programs and structured mobility-enhancing exercises, may contribute to improved health outcomes and enhanced quality of life in this population. Future research should explore the integration of objective physical activity monitoring and predictive model to further refine intervention strategies and ensure their effectiveness in rural settings.

## Supporting information

**S1 File. Supplementary material-appendixes.**
(DOCX)

**S2 File. Data XGBOOST.**
(CSV)

## Acknowledgments

We extend our sincere gratitude to all members of the Alpha Care Team for their invaluable contributions to the organizational design during the onsite survey.

## Author contributions

**Conceptualization:** Xin Zhang, Feng Li, Claudine JC Lamoth.

**Data curation:** Xin Zhang.

**Formal analysis:** Xin Zhang, Claudine JC Lamoth.

**Funding acquisition:** Xin Zhang, Feng Li.

**Investigation:** Xin Zhang, Tianzhuo Yu.

**Methodology:** Xin Zhang, Xiaoping Zheng, Hans Hobbelen, Barbara van Munster, Claudine JC Lamoth.

**Software:** Xin Zhang, Claudine JC Lamoth.

**Validation:** Xin Zhang.

**Visualization:** Xin Zhang, Claudine JC Lamoth.

**Writing – original draft:** Xin Zhang.

**Writing – review & editing:** Xin Zhang, Xiaoping Zheng, Hans Hobbelen, Barbara van Munster, Qian Tong, Tianzhuo Yu, Feng Li, Claudine JC Lamoth.

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
