## [Decision Letter · Decision Letter 0]

23 Aug 2025

PONE-D-25-34585Factors contributing to differences in physical activity levels in (pre)frail older adults living in rural areas of ChinaPLOS ONE

Dear Dr. Zhang,

Thank you for submitting your manuscript to PLOS ONE. After careful consideration, we feel that it has merit but does not fully meet PLOS ONE’s publication criteria as it currently stands. Therefore, we invite you to submit a revised version of the manuscript that addresses the points raised during the review process.

We look forward to receiving your revised manuscript.

Kind regards,

Hidetaka Hamasaki

Academic Editor

PLOS ONE

Journal Requirements:

3. In the online submission form, you indicated that data cannot be shared publicly because of ethical consideration. Data are available from the corresponding author Institutional Data Access / Ethics Committee for researchers who meet the criteria for access to confidential data.

5. Please note that funding information should not appear in any section or other areas of your manuscript. We will only publish funding information present in the Funding Statement section of the online submission form. Please remove any funding-related text from the manuscript.

Reviewers' comments:

Reviewer's Responses to Questions

**Comments to the Author**

1. Is the manuscript technically sound, and do the data support the conclusions?

Reviewer #1: Yes

Reviewer #2: Yes

2. Has the statistical analysis been performed appropriately and rigorously?

Reviewer #1: Yes

Reviewer #2: I Don't Know

3. Have the authors made all data underlying the findings in their manuscript fully available?

Reviewer #1: Yes

Reviewer #2: No

4. Is the manuscript presented in an intelligible fashion and written in standard English?

Reviewer #1: Yes

Reviewer #2: No

5. Review Comments to the Author

Reviewer #1: Introduction:

1.Literature gap justification: The introduction mentions that "it is not yet clear how these factors are related to physical activity levels in pre(frail) older adults especially those living in rural areas." However, could you provide more specific evidence about what aspects of this relationship remain unexplored, given the existing literature on frailty and physical activity?

2.Hypothesis development: The hypothesis states that "self-perceived health will have a major impact on physical activity levels compared to other health-related factors." What theoretical framework or preliminary evidence supports prioritizing self-perceived health over other established predictors like social support or physical performance?

3.Geographic specificity: The introduction emphasizes rural China, but how do the unique characteristics of Northeast China (climate, culture, economic conditions) potentially influence the generalizability of findings to other rural contexts?

Methods

4.Frailty classification consistency: The inclusion criteria require "frailty score >0 according to the Fried Phenotype," but later analysis combines pre-frail and frail participants. How might this heterogeneity within the sample affect the interpretation of physical activity predictors, and should these groups be analyzed separately?

5. IPAQ-SF validity concerns: Given that you acknowledge in the discussion that IPAQ-SF tends to overestimate activity levels with correlations often below acceptable standards, why was this instrument chosen over more objective measures, and how might this limitation affect your primary findings?

6.XGBoost model justification: While you mention XGBoost can handle imbalanced data and non-linear relationships, what specific advantages does this approach offer over traditional logistic regression for this particular research question, especially given the relatively small sample size (n=284)?

Results

7.Class imbalance impact: Despite using class weighting, your model shows 90% specificity but only 63% sensitivity. How does this imbalance affect the practical utility of the model, particularly for identifying individuals who might benefit from physical activity interventions?

8.Feature importance interpretation: The SHAP analysis shows SSRS (social support) as highly influential, but Figure 4 shows considerable overlap between groups. How do you explain this apparent contradiction, and what does this suggest about the model's ability to capture the true relationship?

9. Physical performance paradox: Your results show that even the high physical activity group performed below the 5th percentile for age-matched peers on physical tests. How do you reconcile high self-reported physical activity levels with poor objective physical performance measures?

Discussion

10. Causality concerns: The discussion frequently implies causal relationships (e.g., "self-reported health, social support, and physical activity engagement"), but this is a cross-sectional study. How do you address the potential for reverse causality, where poor physical function might lead to negative health perceptions rather than vice versa?

11.Intervention implications: You suggest "community-based social support programs and structured mobility-enhancing exercises," but given that most participants already met high physical activity levels (despite poor physical performance), what specific interventions would be most appropriate for this population?

12.Methodological limitations: While you acknowledge IPAQ-SF limitations in the discussion, this appears to be a fundamental flaw that may compromise the study's validity. How significantly might measurement error affect your conclusions, and should this limitation be addressed more prominently earlier in the manuscript?

Reviewer #2: This manuscript describes the study of factors contributing to differences in physical activity levels in prefrail older adults in rural areas This is an important topic since frailty prevention in older adults should be a priority in all populations.

General comments:

- Authors should follow financial disclosure guidelines.

- Language revision is required.

- Authors declared all data are available without restrictions, yet they later stated that data cannot be shared publicly. Non-identifiable data can be provided without restrictions, and the statements need to be congruent.

Abstract:

- Abstract sections do not usually include a discussion, instead; they include a conclusion. Please follow Plos One guidelines.

Introduction:

- I am not sure how the Chinese government classifies age groups. Is the definition of older adults starts at the age 60 or 65? The authors included patients who were 60 or above. Yet, in the introduction, the classification states 65 years or above. Please explain why you chose to use the age 60 or more instead of official classification.

- Aim: Authors intended to measure the influence of different health domains on physical activity levels. Since this was a cross-sectional study, causality cannot be proven. All we can confirm here is the presence of an association. I would assume that higher physical activity can positively influence other health domains. This is more logically proven via studies on the positive influences of higher physical activity on many health-related aspects and on diseases.

Methods:

- Including all those with Fried’s criteria scores more than 0 means including patients with frailty as well.

- Please provide the approval to use MMSE as it is copy righted.

- Authors included anxiety, depression and social support under perceived health item in the results. Psychiatric problems are not the only contributors to health and not including physical problems as well can gravely affect the results. Please include other aspects of health in your analysis.

Results:

- Where are the supporting tables? I was not able to find them.

- In table 1: authors did not put the significance of all items. Only significant variables had their p values in the table. Please add all the data in the table.

Discussion:

- Please cite the authors by name when it is part of the sentence structure (Page 25 line 324).

6. PLOS authors have the option to publish the peer review history of their article (what does this mean?). If published, this will include your full peer review and any attached files.

Reviewer #1: No

Reviewer #2: No

---

## [Author Response · Author response to Decision Letter 1]

25 Sep 2025

Please refer to the "response to the reviwer.docx", find the point-to-point answer.

---

## [Editor Report · Decision Letter 1]

14 Oct 2025

Factors contributing to differences in physical activity levels in (pre)frail older adults living in rural areas of China

PONE-D-25-34585R1

Dear Dr. Zhang,

We’re pleased to inform you that your manuscript has been judged scientifically suitable for publication and will be formally accepted for publication once it meets all outstanding technical requirements.

Kind regards,

Hidetaka Hamasaki

Academic Editor

PLOS ONE

Additional Editor Comments (optional):

Thank you for submitting the revised manuscript.

Since there has been no response from the two reviewers, I have checked your responses to the reviewers as the Academic Editor on their behalf. It appears that you have provided careful and detailed replies to the reviewers’ comments and suggestions. Appropriate revisions have also been made to the main text and tables, and I believe the manuscript meets the publication standard.
---

## [Editor Report · Acceptance letter]

PONE-D-25-34585R1

PLOS ONE

Dear Dr. Zhang,

I'm pleased to inform you that your manuscript has been deemed suitable for publication in PLOS ONE. Congratulations! Your manuscript is now being handed over to our production team.

Kind regards,

on behalf of

Dr. Hidetaka Hamasaki

Academic Editor

PLOS ONE